# Policies and Regulations for Desertification Prevention and Control in Mongolia

**Yuan You, Na Zhou and Yongdong Wang \***

National Engineering Technology Research Center for Desert-Oasis Ecological Construction, Xinjiang Institute of Ecology and Geography, Chinese Academy of Sciences, 818 South Beijing Road, Urumqi 830011, China; youyuan@ms.xjb.ac.cn (Y.Y.); zndesert@ms.xjb.ac.cn (N.Z.)

**\*** Correspondence: wangyd@ms.xjb.ac.cn; Tel./Fax: +86-991-782-313-9

**Abstract:** Desertification is a transnational, cross-regional, and global eco-environmental problem that seriously restricts sustainable socioeconomic development. As Mongolia is a typical arid and semi-arid region, the evolution of desertification in the country is closely related to major global issues such as climate change, global carbon cycling, and biodiversity. In this article, we analyze the background, development process, limitations, and other aspects of Mongolia's desertification prevention and control policies and regulations and conclude that Mongolia needs to formulate a "Desertification Prevention and Control Law." In particular, it needs to clarify the responsibility subjects, beneficiaries, interest compensation subjects, and illegal punishment subjects for prevention and control, as well as the responsibilities and obligations of relevant legal subjects. The research results show that it is important to form a solution mechanism in legal research on the joint prevention and control of desertification between Mongolia and China. We propose a concept of best future practice, highlighting the urgent need to establish a framework for the joint prevention and control of desertification via a cooperative mechanism between Mongolia and China and for the two countries to jointly promote global cooperation in combating this important environmental issue.

**Keywords:** Mongolia; desertification; policies and regulations; desertification prevention and control





## 1. Introduction

Desertification is a transnational, cross-regional, and global eco-environmental problem that seriously affects agricultural and forestry production in arid areas and the succession process of desert ecology and severely restricts sustainable socioeconomic development [1–4]. At present, the total desertification area is 3.6 billion hectares globally, accounting for 56.8% of the total land area. Desertification not only affects local residents' livelihoods but also causes land resources to be permanently destroyed, land-resource utilization to be disrupted, and the environment to be seriously damaged [5]. Desertification control is a comprehensive engineering concept involving significant expenses, long timescales, and strong externality across a wide range of fields. Policies, funds, and technology are the basic elements of desertification prevention and control, which, in practical terms, translate first to a reliance on policy development, then investment in policies, and finally the technology to realize those investment-backed policies. Human activities run throughout the whole process of combating desertification [6]. Therefore, it is necessary to regulate the concepts and codes of conduct of the objects of desertification control policy and law; guide, restrain, and manage human activities; increase public awareness of desertification; and encourage public participation in desertification control [7,8]. Desertification control policy and law do not represent a single policy and law system involving all aspects of socioeconomic and environmentally sustainable development; rather, they represent a complex problem that requires the overall consideration of economic interests, environmental and social effects, and their conflicts [9–12].

Mongolia is one of the countries that is most severely affected by desertification. According to the latest data from the United Nations, the desertification land area in Mongolia accounts for 76.9% of the national territory, and desertification is still spreading at a relatively fast pace in some areas [13]. The land area with moderate, severe, and very severe degrees of desertification is expanding. In order to combat desertification, the Mongolian government passed the National Plan to Combat Desertification in 1996 and became a member of the United Nations Convention to Combat Desertification [14]. It also established the National Committee to Combat Desertification. Since 2005, Mongolia has officially implemented the Green Great Wall plan. In 2010, Mongolia's President, Tsakhiagiin Elbegdorj, signed an order to designate the second Saturdays of May and October as "National Tree Planting Days", advocating for active participation in tree planting activities throughout the country to prevent desertification [10,15]. In 2011, the Institute of Geological Environment of the Mongolian Academy of Sciences established the Research Center for Reducing Desertification in Lassaud County, Bulgan Province, to strengthen research and experimental work on reducing desertification. In 2012, the National Bureau for Soil Protection and Desertification Control was established under the Ministry of Natural Environment and Tourism. Over the past decade, the Mongolian government has implemented over 20 environmental protection plans, including the Green Great Wall, Protecting Forests, and Protecting Water plans, to prevent and reduce desertification. There have been approximately 240 projects, with a total investment of MNT 3.5 billion. However, these projects have not yet achieved significant results in preventing desertification [16–18].

Zhao [19] concluded that desertification has seriously restricted the sustainable development of the arid and semi-arid regions in the eastern part of the China–Mongolia–Russia Economic Zone, especially in China and Mongolia. However, his research also showed that, from 2000 to 2020, the degree of desertification in the entire study area has improved, with the area of severe and extremely severe desertification having reduced by 2.23%. Nonetheless, at the national scale, whilst the degree of desertification in China has decreased over time, the area of desertification in Mongolia has continued to expand. Furthermore, Dong [20] observed that desertification risks are mainly concentrated along the cross-border transport belt between China, Mongolia, and Russia. Dong's paper provides scientific and technological support for the rational construction of cross-border railways and pipelines and avoids and guards against ecological risks in the China–Mongolia–Russia economic corridor. This will be the basis for decision-making in promoting green and sustainable development in the region. Meanwhile, Suvd [21] analyzed the relationship between desertification and grassland degradation—specifically, the legal relationship between grassland utilization and management at the national level in Mongolia—and proposed suggestions for modifying grassland protection and grassland-related policies and regulations. Liu et al. [22] analyzed the relationship between desertification and the ecological environment in Mongolia and observed that the main reason for grassland degradation and declines in ecosystem services is the unsustainable management of grasslands. Therefore, they highlighted the need to rebuild the livestock management system, improve the ecological management system, and attach importance to the top-level design of desertification prevention policies and regulations to restore the stability of grassland ecosystems. Liang et al. [15] mentioned that, due to limited understanding of desertification in Mongolia at this stage, the prevention of desertification in Mongolia and the country's sustainable development have both been constrained. They studied the spatiotemporal patterns, driving factors, mitigation strategies, and research proposals in respect of desertification in Mongolia and proposed that existing desertification prevention policies and regulations should be strengthened, especially by considering the relationship between grassland-bearing capacity and animal husbandry development.

Although some scholars, both domestically and internationally, have conducted extensive research on the issue of desertification prevention and control in Mongolia, with different perspectives and methods, there have been very few comparative studies on the formulation of environmental policies and regulations for combating desertification. Cur-

rently, most research is conducted from the perspectives of ecology, the environment, land, forestry, and other disciplines, with relatively few studies on the desertification control policies and regulations in Mongolia. Analyses of the management system for combating desertification have not been sufficiently detailed [23]. From the perspective of the top-level design of desertification control in Mongolia, the aim of this paper is to analyze the background, current situation, and future challenges and problems in the formulation of desertification control policies and regulations.

Mongolia and China are close neighbors, and Mongolia is the main source of sand-storm disasters in East Asia. Indeed, the southern part of Mongolia is a particularly important source of sandstorms in China [24]. Furthermore, as Mongolia is a typical arid and semi-arid region, the evolution of desertification in Mongolia is closely related to major global issues such as climate change, global carbon cycling, and biodiversity [1]. Studies have emphasized that it is important to form a solution mechanism in legal research on the joint prevention and control of desertification between Mongolia and China, jointly promote global cooperation in controlling desertification, strengthen research on regional environmental information reserves and basic scientific research cooperation, and fulfill the obligations of the UNCCD [12,19].

## 2. Study Area

Mongolia is located on the Mongolian Plateau in Northeastern Asia (41°32′–52°15′ N, 87°44′–119°56′ E, Figure 1), with a land area of 1.5665 million km², making it the second largest landlocked country in the world. Mongolia is located in an arid and semi-arid region with low precipitation and a fragile ecological environment [25]. The Köppen climate zone is defined according to the combination of three models: main climate + precipitation + temperature. Mongolia has the following six climate types: Bsk: arid + summer dry + cold arid; Dwb: snow + winter dry + warm summer; BWk: arid + desert + cold arid; Dwc: snow + winter dry + cool summer; Dfc: snow + fully humid + cool summer; ET: polar + polar tundra.

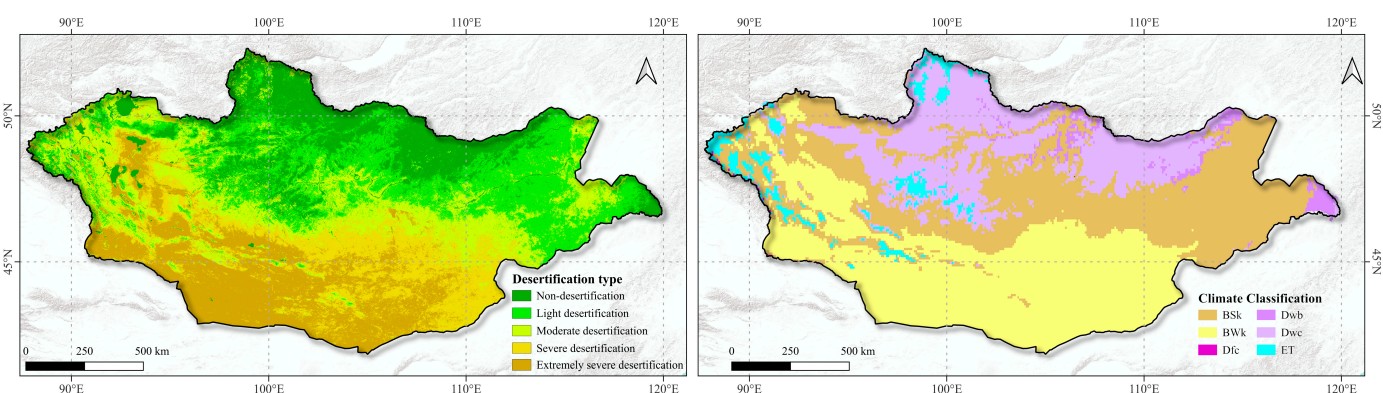

**Figure 1.** Schematic showing the degree of desertification and climatic zones of Mongolia. Sources: https://koeppen-geiger.vu-wien.ac.at/present.htm (accessed on 25 March 2024).

Animal husbandry is Mongolia's leading industry. Due to the announcement in the early 1990s that "grasslands and pastures are owned by the whole people" and "national freedom" in choosing a place of residence, Mongolia's livestock population rapidly increased, ecological and environmental pressures increased, and owing to the impact of climate change, grassland vegetation has declined and desertification has developed rapidly [26]. Under the dual influence of climate change and human economic activities, Mongolia's fragile ecological environment has been greatly affected, and there is an obvious desertification problem marked by vegetation degradation. Desertification has become a major factor restricting the sustainable development of the regional economy and society. Figure 2 shows the spatial distribution of the transformation of the degree of

land desertification in each adjacent period in Mongolia from 1990 to 2020. It can be seen from the figure that the desertification type transformation fluctuates greatly in different periods, and the central region of Mongolia is the region with a high frequency of type transformation in each period; that is, the central region is more sensitive to climate change and human activities. For example, from 2000 to 2005, the desertification type in central and eastern Mongolia mainly showed a degraded change, while from 2005 to 2010, the desertification type in this area showed an obvious escalation change. Although the Mongolian government recognizes the urgency of desertification prevention and control, it is difficult for the government to invest a large budget in ecological construction due to its low national strength, and it is difficult to implement the grand "Grassland Road" plan. Mongolia has become the country most severely affected by desertification in the world and is an important source of sand and dust in Asia, thereby causing the deterioration of the ecological environment in East Asia and even globally and posing a great threat to the sustainable development of the regional economy and society.

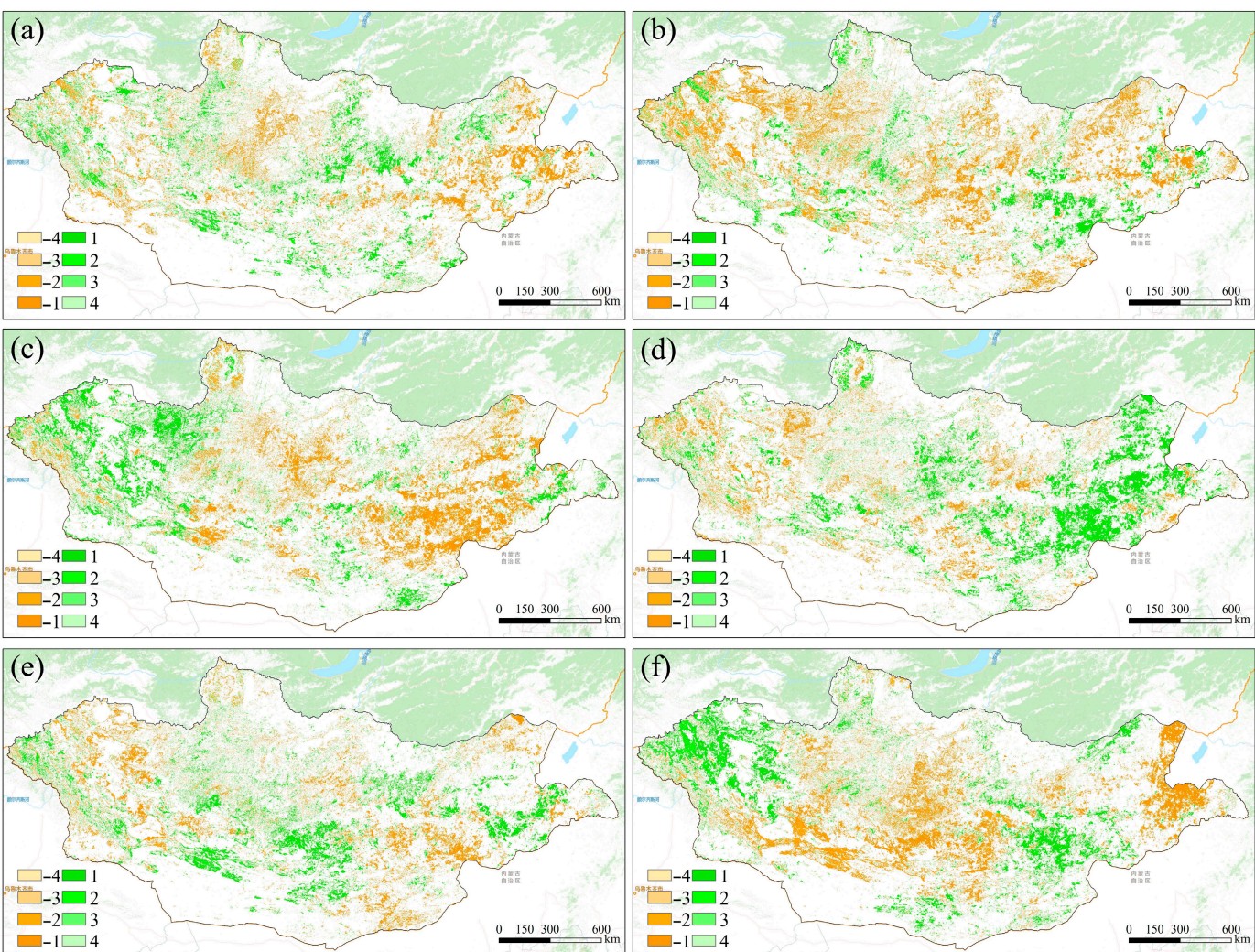

**Figure 2.** Cross-level distribution of desertification degree in Mongolia from 1990 to 2020. Notes: (**a**) 1990–1995; (**b**) 1995–2000; (**c**) 2000–2005; (**d**) 2005–2010; (**e**) 2010–2015; (**f**) 2015–2020. In the legend, −1, −2, −3, and −4 indicate the degrees of desertification intensification, and the smaller the value, the greater the degree of desertification intensification. 1, 2, 3, and 4 indicate the degrees of desertification reduction, and the greater the value, the greater the degree of desertification reduction.

### 3. Analysis of the Process of Formulating Policies and Regulations for Combating Desertification in Mongolia

#### 3.1. Background of Desertification Policies and Regulations in Mongolia

The degree of desertification in Mongolia is becoming increasingly severe. Therefore, Mongolia has implemented a series of national ecological and environmental protection plans, such as the Green Great Wall, Forest Protection, and Water Protection plans, and National Tree Planting Day [17,27]. The development of desertification prevention and control policies and regulations in Mongolia can be divided into three stages in chronological order. These stages are described in detail in Table 1. Briefly, however, the initial stage included accession to the UNCCD in 1996 and the formulation of a national strategy and action plan for desertification prevention and control. Then, in the development stage, in 2011, the National Action Plan for Combating Desertification (2010–2020) was reformulated, and the Research Center for Reducing Desertification was established. Further, a series of laws and regulations were introduced, proposing a "green development policy". Finally, in the epidemic impact stage, since 2020, Mongolia has attached greater importance to aligning with international research on desertification prevention and control and, through international cooperation, better combating desertification [9].

**Table 1.** The stages of development of policies and regulations designed to combat desertification in Mongolia.

| Stage | Specific Policies and Regulations |
|---|---|
| Initial stage (1996–2010) | In 1996, Mongolia joined the UNCCD and formulated a national strategy and action plan for desertification prevention and control. In 2002, the country issued the Land Law. In 2005, Mongolia officially implemented the Green Great Wall plan. In 2007, it promulgated the Plant Protection Law. In 2009, Mongolia's government issued a law prohibiting mineral exploration and mining operations in river headwaters, reservoir protected areas, and forest areas. In 2010, Mongolia's President, Tsakhiagiin Elbegdorj, signed an order to designate the second Saturdays of May and October as National Tree Planting Days. |
| Development stage (2011–2020) | In 2011, Mongolia reformulated the National Action Plan for Combating Desertification (2010–2020), established the Research Center for Reducing Desertification, and formulated the National Climate Change Action Plan. In 2012, the country promulgated the Soil Protection and Desertification Prevention and Control Law [28], the Forestry Law, and Natural Resources Law No. 22; and in 2013, Order A-66 of the Ministry of Environment was introduced, confirming the Regulations on Drought Resistance, Sand Prevention and Control, and Water Resource Management. Also in 2013, Mongolia implemented the Comprehensive Water Management Plan. In 2014, the country implemented the Green Development Policy. In 2015, the National Forestry Policy (2016–2030) was implemented, specifically proposing to increase the forest coverage of Mongolia to 8.3% and 9.0% by 2020 and 2030, respectively [29]. Also in 2015, the Ministry of Nature, Environment, and Tourism issued Order A-138, confirming the Technical and Biological Reclamation Methods for Mining Disturbed Land; plus, the country formulated the Biodiversity Plan. In 2016, Mongolia released its 2030 Sustainable Development Vision. |
| Epidemic impact stage (2020 to date) | In 2020, Mongolia issued the Livestock Tax Law, and the Ministry of Environment and Tourism approved Decree No. A/176 on Forest Ecology and Economic Assessment. In 2022, a Memorandum of Understanding on Forest Partnership was signed with the European Union. |

#### 3.2. Process of Formulating Policies and Regulations for Combating Desertification in Mongolia

Combating desertification is not just a technical issue but also a social and management issue. It involves various aspects of society, economy, and ecology, as well as various government departments such as forestry, agriculture, water conservancy, and environmental protection [10].

"Regulations" is the general term for statutory documents such as laws, regulations, rules, and articles of association. Regulations refer to normative documents formulated by state agencies. Meanwhile, the term "policy" refers to the guiding principles and guidelines for the actions of a country to achieve certain political, economic, cultural, and other goals and tasks. The state formulates policies to clarify the purpose, standards, guidelines, and measures of action. Policy is a product of the development of human society to a certain

stage of class society, with class nature, practicality, universality, scientificity, and flexibility [7]. Environmental policies and regulations represent the extension and concretization of sustainable development strategies and environmental protection strategies and are the concepts and behavioral norms that guide, constrain, and manage the objects of environmental policy regulation. Policies and regulations for combating desertification—namely, laws, regulations, rules, and regulations related to desertification prevention and control, soil and water conservation, and the protection of natural resources and the ecological environment—are used to solve the problems of desertification and the conflicts between its stakeholders.

All data used in this study were downloaded from the National Statistics Office of Mongolia website (https://www.nso.mn/en accessed on 7 March 2024) and the UN Statistics Division (https://unstats.un.org/UNSDWebsite accessed on 7 March 2024).

### 3.2.1. The Intensification of Land Desertification Caused by Animal Husbandry

In 1991, Mongolia promulgated the Mongolian Property Privatization Law. Then, in 1992, the privatization of agriculture and animal husbandry was completed, with 90% of livestock and land privately owned. Mongolia has implemented a pastoral management model of "grass pastures are owned by the whole people" and "national freedom" in choosing a place of residence while livestock are privately owned, resulting in a contradiction between public ownership of grass pastures and private ownership of livestock [26]. Due to overgrazing, grassland resources and the ecological environment have been greatly damaged, leading to the tragedy of the commons and accelerating regional desertification [27].

Unlike large-scale, intensive animal husbandry countries, Mongolia still maintains nomadic production and a lifestyle that involves living near water and grass. According to data from the National Bureau of Statistics of Mongolia, during the planned economic period from 1970 to 1990, the livestock in Mongolia remained between 20 million and 25 million. Since the implementation of privatization in 1992, the number of livestock on hand has increased significantly. In 1997, it exceeded 30 million; in 2007, it exceeded 40 million; in 2010, affected by the "white disaster" (snow disaster), it fell back to 32.73 million, and then continued to grow for nine consecutive years, reaching 70.97 million in 2019; and in 2020, affected by the COVID-19 pandemic, it fell back to a total of 67.07 million [30]. In Mongolia, grazing is allowed in all seasons except for winter, when grass cannot be eaten after snow. Livestock can eat hay even in winter when there is no snow cover [31]. The livestock number of 70.97 million in 2019 was accompanied by a suitable grassland grazing amount of 51.63 million, and the overgrazing amount was 19.34 million, with an overgrazing rate of 37% (Figure 3). Compared to an increase in the stock, changes in the proportions of different types of livestock within the overall structure have a greater impact on the grassland. In the long term, Mongolia has formed a harmonious and symbiotic "five livestock" structure (namely, horses, cows, camels, sheep, and goats) based on the feeding characteristics of different animal species. However, due to the market economy and national policies (the Mongolian government's passing of the National Cashmere Plan in 2018), the stock of goats has surged and is basically now equivalent to the number of sheep [30]. The increasing number of goats poses a challenge to the environmental carrying capacity, as the grassland eaten by goats and sheep takes a considerable amount of time to recover. Such an imbalance in the livestock structure leads to the reverse succession of grassland ecology, exacerbating the problem of desertification. In addition, after the privatization of livestock, Mongolia did not carry out effective changes in the property rights system in respect of grasslands, resulting in unplanned use of grasslands, which is an important factor in exacerbating desertification [27].

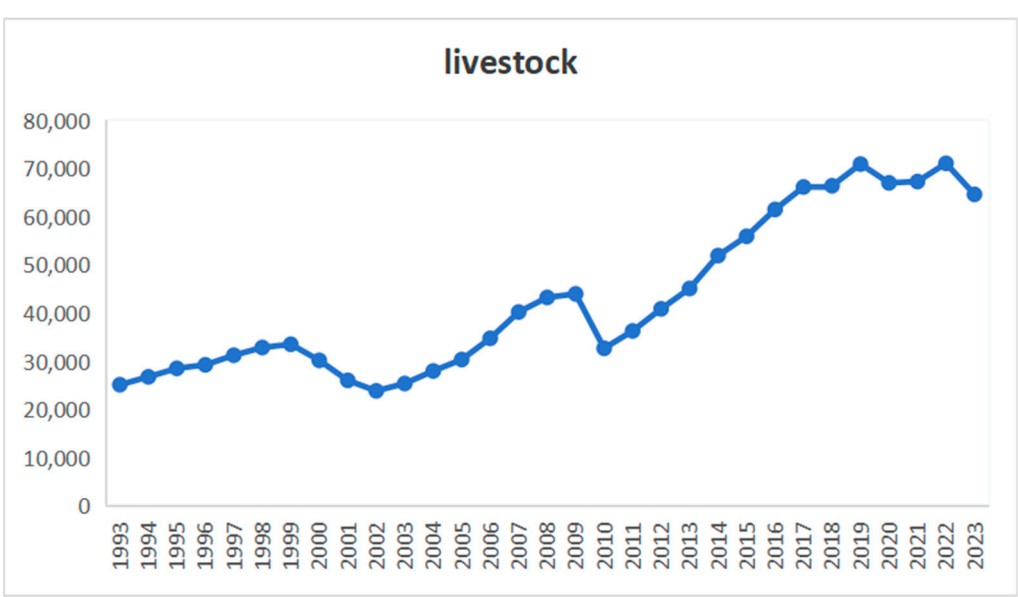

**Figure 3.** Changes in livestock numbers in Mongolia.

3.2.2. The Intensification of Land Desertification Caused by the Mining Industry

Since entering the 21st century, driven by the market economy and national policies, the mining industry has experienced leapfrog development, which is another human factor that cannot be ignored in the intensification of desertification in Mongolia.

In the early 21st century, driven by the strategy of "mining to revitalize the country," Mongolia's mining and export of gold, copper, lead, zinc, coal, and other mineral products increased year by year. Taking coal exports as an example, the total export volume in 2003 was only 1.63 million tons, but by 2022, it had reached 27.1059 million tons—a 17-fold increase in 19 years. In a short period of time, a large number of mining activities have caused varying degrees of damage to the ecological environment, resulting in rivers drying up, groundwater levels dropping, and soil erosion, all of which have contributed to the acceleration of land desertification. During the process of mineral extraction, a large amount of soil accumulation increases the source and flow of sand dust, thereby exacerbating grassland desertification. In addition, due to the substantial water resources required for mineral extraction, the demand for water resources in Mongolia's mining industry is also increasing. For example, during the period from 2004 to 2006, gold mining used 76.8 million cubic meters of water, 96.8 million cubic meters of water, and 93.8 million cubic meters of water, respectively [32]. According to data released by the Ministry of Natural Environment and Tourism of Mongolia, since 2010, a total of 1244 rivers and lakes have dried up and stopped flowing [33,34]. As of 2015, mining had directly caused 27,070 hectares of soil erosion and land degradation. The ecology of grasslands is already fragile, and large-scale mining can result in the permanent disappearance of grasslands. Improper treatment of tailings also puts greater pressure on the environment. Various vicious cycles have led to increasingly serious environmental problems in Mongolia.

*3.3. Limitations of the Existing Policy and Regulatory System*

The Mongolian government is not fully aware that desertification and land degradation are the most serious ecological threats facing Mongolia; these will have a negative impact on the sustainable development of society and the economy. Insufficient understanding leads to insufficient attention. The government-led desertification prevention and control work presents a phased characteristic, and in terms of finance, it is mainly assisted by international organizations, supplemented by government budget investment [18,23]. The availability of funds directly affects the implementation of desertification prevention and control work, as the prevention and control measures are single and not comprehen-

sive. The main method for preventing desertification, reducing land degradation, and lessening the impacts of frequent sandstorms is to plant trees and grass to increase the vegetation coverage. According to the budget, the annual planted area of the country is 3000–5000 hectares, while the "Sustainable Development in Mongolia—2030" policy clearly states that the forest coverage rate should reach 8.5% from 2016 to 2020, 8.7% from 2021 to 2025, and 9% from 2026 to 2030. Under the framework of the United Nations' 2030 Ecological Restoration Plan, the 2020 United Nations Convention to Combat Desertification provided $560,000 in assistance to Mongolia to implement tree planting and sand fixation work in Zamyn Uud, Donggobi Province. In fact, the current level of vegetation coverage in Mongolia only accounts for 7.9% of the national territory. The goal of controlling desertification is to restore the ecosystems lost to it [35], but the effectiveness of such restoration work depends on policy support, technical support, and human resource guarantees, which are expensive. From the perspective of the current national conditions in Mongolia, the aforementioned support and guarantees are presently not available, making it difficult for restoration work to achieve results and the established goals.

Since the 1990s, Mongolia has implemented a series of policies and measures to alleviate desertification. For example, in 1996, Mongolia joined the UNCCD and implemented the National Desertification Control Plan. In 2005, the National Committee for Desertification Prevention and Control was established, the Green Great Wall Plan was launched, and since 2008, efforts have been made to strengthen land reclamation in mining areas. However, due to inadequate funding and a lack of continuity in project implementation, the effectiveness of desertification prevention and control has been limited. For example, the Mongolian Green Great Wall Project (also known as the Green Belt Ecological Belt National Plan) aims to establish approximately 3000 km of green belts in southern Mongolia. However, the project lacked prior environmental impact assessment and cost–benefit analysis, top-level design solutions, and scientific on-site environmental research. Therefore, it was implemented without good planning and management, and ultimately, the survival rate of planted seedlings has been very low, and the expected goal to effectively curb the expansion of desertification has not been achieved [36].

The Mongolian government has made some changes to its policies to combat desertification, mainly influenced by the announcement of the UNCCD and the adoption of a new strategy for the next decade by the UNCCD in 2008. The Mongolian government is also attempting to incorporate desertification prevention into its departmental policies. The prevention and control of desertification can only be implemented through strong, coherent, and transparent policies, not only in the environmental field but also in other sectors. However, this issue is not fully emphasized in policy documents. The Law on Protecting Soil and Preventing Desertification, issued in 2012, is a separate law in Mongolia to prevent land desertification. Its special feature is the establishment of specialized agencies to manage the prevention and control of desertification. This law makes clear provisions regarding the levels of soil degradation and desertification, the measures for protecting soil and preventing desertification, the powers of national agencies in protecting soil and preventing desertification, the rights and obligations of citizens and legal professionals in protecting soil and preventing desertification, and the reward system for repairing land desertification [19,37]. However, regardless of the legalities or specific provisions, this law focuses on the prevention of land desertification and has few relevant provisions regarding the control of land desertification. In addition, the provisions with respect to preventing land desertification in the Law on the Protection of Soil and the Prevention of Desertification are too general. For example, the general measures for desertification prevention and control in Article 6.2 are as follows: establishing a sustainable land management system based on regional characteristics, encouraging the development of isolated forest belt planting, rational utilization, protection of and an increase in water resources, protecting forests and plants, planting trees, and protecting the soil of planted agricultural land; while Article 11.3.6 stipulates measures for protecting soil and preventing desertification, which are only divided into protecting and restoring soil, preventing desertification, alleviating

desertification, and combating desertification, without specific action measures (such as how to do it, where to do it, etc.).

The government of Mongolia has issued several governance programs in recent years (within the last decade) without specifically addressing the prevention and control of desertification. In order to develop the economy, the Mongolian government passed the National Cashmere Plan and Three Pillar Economic Development Policies in 2018, vigorously promoting the development of two major foreign exchange-earning industries—the cashmere industry and the mining industry. In 2021, the production of cashmere in Mongolia reached 9700 tons, while the position of mining in Mongolia's national economy has become increasingly important, with its proportion of gross domestic product (GDP) increasing from 20.1% in 2016 to 25% in 2022 [36].

## 4. Discussion

In order to improve the effectiveness of desertification control in Mongolia, scholars have proposed more sustainable prevention measures. For example, in grasslands, appropriate grazing is advocated to avoid overgrazing and alleviate grassland degradation and desertification. The recommended time for grazing activities in campsites is 15–20 days; some studies have also proposed adjustments to the grazing system and traditional nomadic feeding methods. In terms of government policy supervision, it is recommended that investment be strengthened, as well as the guidance and supervisory role of the government. These suggestions also include (1) improving the legal system, (2) formulating and promulgating relevant land protection policies and land management laws, and (3) implementing mandatory measures to control land degradation. In addition, some other relevant suggestions include (1) improving the social service system in pastoral areas, such as effectively utilizing idle pastures to reduce the pressure on pastures in overgrazed areas; (2) vigorously developing the deep processing industry of livestock products; (3) improving the upstream and downstream industrial chain; and (4) expanding the income sources of herdsmen and other agricultural workers.

We believe that the national action plans, policies, and regulations for combating desertification should be a sustainable operational mechanism for the interaction between agriculture, animal husbandry, forestry, water resource management, and industry. The causes of desertification are closely related to human activities and socioeconomic processes, and the ultimate practical direction of desertification control policy formulation is to alleviate the pressure on the ecological environment and achieve sustainable socioeconomic development.

Although Mongolia's economy has been developing rapidly, due to weak infrastructure and technical forces, an insufficient economic structure, and a structural lack of human resources, the overall economic volume is small and the poverty rate remains high; the economy is largely dependent on external factors, and is greatly affected by fluctuations in the international financial market; the government has a serious debt burden and a serious shortage of funds; and the overall social ability to prevent and control desertification is weak. Therefore, although the Mongolian government realizes the urgency of desertification prevention and control, certain national weaknesses mean that it is difficult for the government to provide a large enough budget for investment in ecological construction, and the grand Steppe Road plan is difficult to implement. The formulation and implementation of policies and regulations for desertification prevention and control are also insufficient, leading to the occurrence of illegal logging and grazing by some enterprises and individuals. This requires combining the national conditions of Mongolia, inheriting the traditional pasture management system in semi-arid and arid areas of Mongolia, and rebuilding the livestock management system to restore the stability of the grassland ecosystem. In summary, the following three specific actions are proposed:

(1) Establish a top-level design for formulating desertification prevention and control policies and regulations.

When it comes to desertification prevention and control policies, the top-level design of policy formulation should be based on solid investigation and research, with a deep and thorough understanding of the actual specific situation of the target areas where the policies are applicable. Implementation should focus on whether the policy is coordinated with the natural conditions of the place of execution and whether it is suitable for that location's economic situation. Reasonable adjustments should be made within the scope allowed by the policy objectives, tailored to local conditions, so that the policy's implementation is compatible with the local natural and economic conditions, truly promoting the improvement of the quality of life of local residents while protecting the natural environment.

Thus far, Mongolia has implemented a series of policy measures to combat desertification. However, some projects have lacked planning, management, and financial support, and the expected effects of desertification prevention and control have not yet been achieved. Due to the very low population density in Mongolia, the majority of people live in rural areas outside of cities. This has caused great difficulties for Mongolia in implementing policies such as desertification prevention and land greening. Policy designers and managers need to take into account the actual situation in Mongolia, and when formulating desertification prevention and control policies, they should fully consider the relationship between grassland changes and the development of animal husbandry, as well as the factors of pasture carrying capacity and pastoralist livelihoods, in order to achieve a balance between grassland protection and animal husbandry, promote precise management models for pastoralists, and encourage the integration of agriculture and animal husbandry.

The quantity of policies and laws does not constitute a principal factor at all, and this quantity is only the final manifestation. It cannot measure the value of any dimension; it can only be the result of one being presented. A country's policy and regulatory system should be as complete as possible, manifested with more details, more perspectives, more comprehensive bills, and more rigorous thinking.

(2) Implement desertification prevention policies and regulations that consider sustainable development issues.

Land desertification in Mongolia is not only an environmental issue but also closely related to the country's economic and social development. Taking the important traditional industry of animal husbandry in Mongolia as an example, overloading and overgrazing of livestock have become the main factors causing grassland degradation in Mongolia. As a leading industry in Mongolia, animal husbandry occupies a significant position in the national economy. Therefore, the utilization of grasslands is significant, and more attention should be paid to their protection and restoration. Mongolia lacks specialized legislation for grassland protection and restoration, and the legal basis for grassland protection is scattered in legal documents such as the Natural Environment Protection Law and the Land Law.

If we want to fundamentally improve the severe situation faced by Mongolia's ecological environment, the joint prevention and control of desertification between China and Mongolia needs to be linked to the sustainable development of Mongolia's economy, and we need to adopt an "ecological protection + industrial development" model. Mongolia is currently actively accelerating its economic transformation. For example, in 2022, Mongolia signed a Memorandum of Understanding with the European Union on the Forest Partnership, which includes creating employment opportunities and promoting socioeconomic development through a sustainable forest value chain. The cooperation between China and Mongolia in desertification prevention and control should also fully consider the balance between ecological protection and Mongolia's economic development, and include promoting, through cooperation the sustainable development of Mongolia's animal husbandry and ecological sand industry.

(3) Optimize desertification prevention policies and regulations through strengthening international cooperation.

The Mongolian government needs to strengthen cooperation with neighboring countries to address environmental issues such as desertification. Desertification prevention

and control represent a huge and complex system-engineering problem that not only requires cooperation from multiple disciplines and departments but also draws on successful experiences and lessons from the failures of developed countries.

We believe that the similarity and specificity of policies and regulations are variable. Therefore, the process of improving and supplementing policies and regulations for combating desertification in Mongolia must learn from the advanced policies and regulations of different countries, regions, and ethnic groups, and use multiple perspectives and dimensions to measure the complexity of desertification, thereby further enhancing the fairness, timeliness, effectiveness, and enforceability of laws. Such an approach is conducive to formulating and improving diversified desertification policies and regulations required by different regions.

## 5. Conclusions

The main conclusion of this paper is that policies and regulations are the most important factors in desertification prevention and control in Mongolia, and research on desertification prevention and control from a policy perspective is urgent and necessary. Desertification control requires the establishment of a comprehensive policy and legal system, guided by policies and regulations, in order to achieve stability and progress on the road to desertification control. Mongolia needs to enact a "Desertification Prevention and Control Law", especially to clarify the responsibilities, beneficiaries, compensation, and punishment of subjects for prevention and control. A good ecological environment is central to the welfare of people and their livelihoods, but it also needs to be protected by a rigorous system and strict rules of law, as well as coordinated with global ecological civilization.

**Author Contributions:** N.Z. and Y.W. designed the study, and collected and analyzed the data. Y.Y. wrote the original manuscript, with all authors contributing towards finalizing the manuscript. All authors have read and agreed to the published version of the manuscript.

**Funding:** This research was funded by the young scientist fund of NSFC "Study on the differences of policies and regulations on combating desertification in 'The Belt and Road Initiatives'" (42207549) and the "Double track implementation mechanism for combating desertification in China and the experiences-sharing in the affected countries along the Belt and Road region" (ANSO-SBA-2021-06).

**Institutional Review Board Statement:** Not applicable.

**Informed Consent Statement:** Not applicable.

**Data Availability Statement:** All data used in this study were downloaded from the National Statistics Office of Mongolia website (https://www.nso.mn/en accessed on 7 March 2024) and the UN Statistics Division (https://unstats.un.org/UNSDWebsite accessed on 7 March 2024).

**Acknowledgments:** This research was supported by the Alliance of International Science Organizations of Association (NSO-PA-2023-15).

**Conflicts of Interest:** The authors declare no conflicts of interests.

## Abbreviations

UNCCD: United Nations Convention to Combat Desertification.

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
