# Peer review of "Policies and Regulations for Desertification Prevention and Control in Mongolia"

_land, doi:10.3390/land13040559_

Round 1

Reviewer 1 Report

Comments and Suggestions for Authors

This article is not acceptable in its current state. Regarding the issue of desertification, it is necessary to provide more information about the factors affecting desertification in Mongolia from the past to the present. The challenges are further explained. To what extent the proposed policies and programs can reduce the intensity of desertification.

In the study area section: Please write the climate characteristics of Mongolia. You can show the climatic zones of Mongolia on a map.

In line 211-212: Is the grazing capacity determined for the number of livestock in Mongolia?  In other words, how many animals and how many months of the year can use Mongolian rangelands

Is it true (line 225-226)? How much water have these mines consumed? Do only mines cause the reduction of underground water and the drying up of rivers? Does agriculture have no effect? These items should be reviewed and analyzed in your article.

 In line (228-229) : Are mines the reason for the drying up of so many rivers and lakes? Sufficient documents must be provided. Have climate changes contributed to the drying up of these lakes and rivers?

Do all rivers and lakes dry up due to the mining? Sufficient documents must be presented. Have climate changes not affected the drying up of these lakes and rivers?

These are generalizations and are not enough, the effects of these methods and suggestions should be quantified, how was it before and how will it be in the future? Even if these suggestions have not yet been implemented. At least the conditions before the implementation of this method should be stated so that it can be compared with the implementation of the new method presented..

It is necessary to write a more comprehensive analysis of the existing challenges before the result. The content you have already written is similar to a report and is not enough for problems.

Author Response

Thank you for your advice. We agree with the reviewer and we gave the native speaker, to improve grammatical and stylistic mistake.

  • Werewrite the study area section, and write the climate characteristics of Mongolia. We have also revised the map of the study area to add climate zones for Mongolia. We also added “Cross-level distribution of desertification degree in Mongolia from 1990 to 2020”, Figure 2 shows the spatial distribution of the transformation of land desertification degree in each adjacent period in Mongolia from 1990 to 2020. It can be seen from the figure that the type transformation of desertification fluctuates greatly in different periods, and the central region of Mongolia is the region with high frequency of type transformation in each period, that is, the central region is more sensitive to climate change and human activities. For example, from 2000 to 2005, the desertification type in central and eastern Mongolia mainly showed a degraded change, while from 2005 to 2010, the desertification type in this area showed an obvious escalation change. Although the Mongolian government recognizes the urgency of desertification prevention and control, it is difficult for the government to invest a large budget for ecological construction due to its weak national strength, and it is difficult to implement the grand "grassland Road" plan.

  • Grazing capacity is not determined by the number of livestock in Mongolia.We re-explained the intensification of land desertification caused by animal husbandry. Grazing in Mongolia is supposed to be allowed except in winter when it is not possible to eat grass after the snow, and livestock can eat hay even in winter when there is no snow cover.

  • Thank you for the reviewer's suggestion. This is true. The grassland ecology in Mongolia is already fragile, and large-scale mining means the permanent disappearance of grasslands. Some projects are transported by road to ports, which will also cause great destruction to the grassland environment in the way. What's more, mining causes water resources to be "overdeveloped". A large number of mines are polluting and depleting the water resources that people and animals rely on for survival. According to the World Bank, due to large-scale mining, the groundwater in Mongolia's South Gobi province can only last for about 10 years, causing droughts and threatening other ecological environments. Mongolia's agricultural system is extremely fragile. So far, it is still mainly nomadic, which makes the country more vulnerable to famine in the face of natural disasters. The instability of agriculture is also the main reason why Mongolia's population cannot grow. The current agricultural situation can barely feed 3 million people, and any further growth will bring great danger. These have been mentioned in the "Study area".

  • We have added the references that "a large number of mining activities in a short period of time directly lead to river drying and flow suspension, causing serious damage to the ecological environment":
  1. McIntyre, N.; Bulovic, N.; Cane, I.; et al. A multi-disciplinary approach to understanding the impacts of mines on traditional uses of water in Northern Mongolia. Total Environ. Sci. 2016, 557, 404-414. https://doi.org/10.1016/j.scitotenv.2016.03.092
  2. Tao,S.; Fang, J.; Zhao, X.; et al. Rapid loss of lakes on the Mongolian Plateau. P. Natl A Sci. 2015, 112, 2281-2286.
  • The reviewer's opinion is very correct. The comparison of policies and regulations before and after implementation is a very effective quantitative study. However, the author is very sorry that this comparative quantitative study cannot be carried out at present because of the difficulty in obtaining data in Mongolia and the incomplete data preservation. However, the author has contacted Khaulanbyek Akhmadi, director of the Land Use and Urbanization Research Office of the Institute of Geography and Geoecology of the Mongolian Academy of Sciences, who is willing to help us. He also mentioned the difficulty and time required for data acquisition. It is hoped that the author can improve this part in the future.

Reviewer 2 Report

Comments and Suggestions for Authors

This manuscript presents a Study on the "Research on Policies and Regulations for Desertification Prevention and Control in Mongolia: A case study of the Lanzhou-Xining Urban Agglomerations.". This is an interesting study that aligns well with the scope and topics of the Journal. There are, however, many inconsistencies within the manuscript that must be clarified prior to acceptance for publication can be recommended.

1. What is the originality of this study?

2. Please emphasize the research gap in this study.

3. "Figure 1. Schematic of the degree of desertification in Mongolia." More details are required. 

If the authors use this map for analysis, further clarification and additional information are needed regarding this map. 

4. Please clearly mention the material and methods (data, references..etc). This will help readers understand the study.

5. The discussion section should be enhanced (discussion is needed comparing other related studies).

6. The reference style should be corrected.

References

References must be numbered in order of appearance in the text (including citations in tables and legends) and listed individually at the end of the manuscript. We recommend preparing the references with a bibliography software package, such as EndNote, Reference Manager or Zotero to avoid typing mistakes and duplicated references. Include the digital object identifier (DOI) for all references where available.

Citations and references in the Supplementary Materials are permitted provided that they also appear in the reference list here.

In the text, reference numbers should be placed in square brackets [ ] and placed before the punctuation; for example [1], [1–3] or [1,3]. For embedded citations in the text with pagination, use both parentheses and brackets to indicate the reference number and page numbers; for example [5] (p. 10), or [6] (pp. 101–105)."

Comments on the Quality of English Language

Minor editing of English language required.

Author Response

We thank the reviewer for constructive criticism. We agree with the reviewer and we reformatted the references.

  • Werevised the introduction.

  • This article analyzes the background, development process, limitations, and other aspects of Mongolia’s desertification prevention and control policies and regulations, and concludes that Mongolia needs to formulate “Desertification Prevention and Control Law”. In particular, it needs to clarify the responsibility subjects, beneficiaries, interest compensation subjects, and illegal punishment subjects for prevention and control, as well as the responsibilities and obligations of relevant legal subjects. Research results show that it is very important to form a solution mechanism in legal research on the joint prevention and control of desertification between Mongolia and China. This paper puts forward the idea of best future practice, highlighting the urgent need to establish a framework for the joint prevention and control of desertification via a cooperative mechanism between Mongolia and China, and for the two countries to jointly promote global cooperation in combating this important environmental issue.

  • We added “Cross-level distribution of desertification degree in Mongolia from 1990 to 2020”, Figure 2 shows the spatial distribution of the transformation of land desertification degree in each adjacent period in Mongolia from 1990 to 2020. It can be seen from the figure that the type transformation of desertification fluctuates greatly in different periods, and the central region of Mongolia is the region with high frequency of type transformation in each period, that is, the central region is more sensitive to climate change and human activities. For example, from 2000 to 2005, the desertification type in central and eastern Mongolia mainly showed a degraded change, while from 2005 to 2010, the desertification type in this area showed an obvious escalation change. Although the Mongolian government recognizes the urgency of desertification prevention and control, it is difficult for the government to invest a large budget for ecological construction due to its weak national strength, and it is difficult to implement the grand "grassland Road" plan.

  • We revised the method.

  • We revised the discussion.

  • We reformatted the references.

Reviewer 3 Report

Comments and Suggestions for Authors

It is important to consider whether the citation format is allowed in the journal

From the introduction, general and short, they  move on to the study area and the analysis of results without mentioning the methodology.

It is also important to clarify the objective and research questions in the Introduction.

Each figure must have the legend on each axis, what the X and Y axis mean In the discussion, the results should be compared with other studies.

Some policy should be suggested based on the results.

Comments on the Quality of English Language

The document dont need English Edit

Author Response

We thank the reviewer for constructive criticism. We agree with the reviewer and we reformatted the references.

We revised the structure of the entire paper, and the whole paper has been modified according to the reviewer's opinion. We gave the native speaker, to improve grammatical and stylistic mistake.

We appreciate the comments from the reviewers. Thank you for reviewing our manuscript.

Round 2

Reviewer 1 Report

Comments and Suggestions for Authors

I wish the finalized article would be sent along with the highlighted article. Some items and suggestions in the article have been modified or completed.

Some limitations to provide the required data, especially for the last question, are explained by the author.

It is suggested that the explanations related to the effects of mines on the drying up of wetlands and rivers and desertification which explained in the referee's response, be written in the article.

If possible, the discussion and conclusion section be enriched.

Author Response

We rewrite 3.2.2 section, “During the process of mineral extraction, a large amount of soil accumulation increases the source and flow of sand dust, thereby exacerbating grassland desertification. In addition, due to the substantial water resources required for mineral extraction, the demand for water resources in Mongolia's mining industry is also increasing. For example, during the period from 2004 to 2006, gold mining used 76.8 million cubic meters of water, 96.8 million cubic meters of water, and 93.8 million cubic meters of water respectively”.

We have added the references:

  1. Bu, R.G.W. Study on the current status, causes and progress of desertification in Mongolia. Master’s Thesis of Inner Mongolia University, 2011, Inner Mongolia. (In Chinese)

Reviewer 2 Report

Comments and Suggestions for Authors

I accept the current version of the manuscript.

Author Response

Thank you for your advice. We agree with the reviewer and we gave the native speaker, to improve grammatical and stylistic mistake.
